# Study on Multi-Step Creep Aging Behavior of Al-Li-S4 Alloy

**Yunlong Ma** [1,2,3], **Feng Xia** [1,3], **Lihua Zhan** [1,3,*] and **Yongqian Xu** [1,3,*]

1   School of Mechanical and Electrical Engineering, Central South University, Changsha 410083, China
2   Beijing Institute of Aerospace Systems Engineering, Beijing 100076, China
3   State Key Laboratory of High-performance Complex Manufacturing, Central South University, Changsha 410083, China
*   Correspondence: yjs-cast@csu.edu.cn (L.Z.); yongqian.xu@csu.edu.cn (Y.X.); Tel.: +86-0731-8887-9351 (L.Z.); +86-0731-8883-0254 (Y.X.)

**Abstract:** Creep age forming (CAF) is a new technology developed for manufacturing large aluminum components in the aerospace industry. Aluminum–lithium alloys may be used in aerospace components because of their high modulus, specific strength and specific stiffness. Therefore, the creep deformation, mechanical properties and aging precipitation of Al-Li-S4 alloy under CAF conditions were studied. It was found that the creep behavior presents double steady state creep stages during the creep aging process. With the increase of stress level, the first steady creep rate increased, but the second steady creep rate was slightly reduced. Coincidentally, in the first steady state creep stage, the yield strength of the studied alloy also showed a slow increase stage. TEM observation showed that Al-Li-S4 alloy mainly contains two precipitation phases, $T_1$ phase and $\theta'$ phase. A few precipitates form during the first steady creep stage. Then, a lot of nucleation and growth of $T_1$ phase resulted in rapid increase of yield strength. At the same time, the increase of stress level effectively inhibited the growth of $T_1$ phase, which resulted in these strengthening phases being more uniform, and thus improved the mechanical properties of materials. On this basis, the relationship between the multi-step behaviors of creep, mechanical properties and aging precipitates are discussed. It is considered that the main reasons for the multi-step phenomenon of creep and mechanical properties are strongly related to the nucleation, growth and distribution of $T_1$ phase.

**Keywords:** Al-Li-S4 alloy; creep aging; multi-step phenomenon; mechanical properties; distribution characteristics

## 1. Introduction

Creep Ageing Forming (CAF) is a metal forming technology developed for manufacturing large aluminum components in the aerospace industry [1]. In recent years, Al-Li alloys have attracted much attention because of low weight, high stiffness and high strength for their applications in aircraft, aerospace and military fields [2–5]. Combining creep aging with Al-Li alloy, it is necessary to explore the influence of different creep aging parameters on the forming of Al-Li alloy. Hu et al. [6,7] studied the effects of creep aging and single stress-free aging on the mechanical properties and micro-precipitates of Al-Li-S4 alloy, and then studied the effects of pre-deformation on creep aging and related mechanical properties of Al-Li-S4 alloy. For creep aging forming process, many scholars have developed a special unified field constitutive model for CAF process. Sallsh et al. [8] divided the aging forming process into three stages by establishing the constitutive equation of material aging forming and use this model to simulate the forming process. The experimental data are basically consistent with the simulation prediction results. Kowalewski et al. [9] used hyperbolic sinusoidal function to describe the constitutive

model for the first time. By introducing three state variables into the model, the whole creep aging process can be expressed more accurately. Huang et al. [10] noticed that the third stage of creep usually does not need to occur in practical applications, so the Kowalewski model is further simplified to reduce the impact of the third stage of creep in order to meet the requirements of practical application. The aging experiments of 7075 aluminum alloy were carried out by Narimetla et al. [11]. The creep aging process of aircraft panel was simulated by the finite element method. The nonlinear Maxwell viscoelastic stress relaxation model is applied to the simulation process, and the aging process is divided into three stages: loading, holding and unloading. Zhan et al. [12] introduced dislocation strengthening into the constitutive model for the first time by establishing a unified creep aging constitutive model of 7055 aluminum alloy, which explained the aging strengthening mechanism of the alloy more completely and was well verified on the aluminum alloy. Through ABAQUS finite element analysis software, Li et al. [13] analyzed the influence of temperature and other parameters on the resilience of stress relaxation of a panel. A constitutive model for 7050 aluminum alloy in pre-aging state and for finite element analysis was established. Experiments show that the stress relaxation becomes more obvious with the increase of temperature, and the rebound of components decreases, which means that the forming accuracy of the component is improved. Lam et al. [14] Zhan's constitutive model was applied to the finite element analysis of creep aging of AA 2219 alloy, and the springback of CA 2219 alloy in CAF process was successfully predicted. Zhan et al. [15] established the stress relaxation constitutive equation of 2219 aluminum alloy is by comparing the difference of creep behavior and stress relaxation behavior of 2219 aluminum alloy, based on creep theory. The material constants of the constitutive model are obtained by regression fitting, but the model can only describe the stress relaxation behavior under a single stress, which has certain limitations. At present, the constitutive model of Al-Li alloys is relatively few, especially for the "multi-step" creep characteristics of Al-Li alloys.

The creep aging behavior of Al-Li-S4 aluminum alloy under different applied stress conditions and its macroscopic model are studied in this paper. The creep aging temperature is 153 °C, the creep time is 5, 10, 15, 20 and 25 h, and the creep stress is 200, 220 and 240 MPa. Firstly, creep aging behavior of Al-Li-S4 alloy is analyzed from the point of creep mechanism. The micro-structure of Al-Li-S4 alloy during creep aging process is analyzed by using Titan G2 60-300 spherical aberration correction projection electron microscopy(FEI Company, Hillsboro, OR, USA) to photograph the high-angle annular dark field (HAADF) inside the alloy and energy spectrum analysis (EDS), as well as the tensile tests of creep specimens were carried out on CMT-5105 electronic universal testing machine to analyze(SUST Company, Zhuhai, China) the microstructural evolution and mechanical properties of the alloy during creep aging.

## 2. Materials and Methods

### 2.1. Materials and Programmes

In this paper, the material is the third generation Al-Li alloy, the grade is Al-Li-S4 Al alloy, and the material is the initial T8 plate. The chemical composition of the alloy is shown in Table 1.

**Table 1.** Chemical composition table of Al-Li-S4 aluminum alloy (mass percentage, %).

| Element | Cu | Mg | Mn | Fe | Si | Zn | Zr | Li | Ti | Al |
|---------|------|------|------|-------|-------|------|------|------|-------|------|
| Concentration | 3.64 | 0.71 | 0.29 | 0.028 | 0.014 | 0.36 | 0.12 | 0.69 | 0.026 | Bal. |

Since the initial material is Al-Li-S4 aluminum alloy T8 state plate, the artificial materials are first solution treated before artificial aging and creep aging, so that the original precipitation phase is re-dissolved into the Al matrix. In this paper, the creep aging test is carried out under different applied stress conditions, and the creep behavior characteristics are analyzed, and the constitutive equation is established. According to the clamp size requirement of RDL50 creep (SUST Company, Zhuhai, China),

the creep test samples with the gauge length of 50 mm were machined by the wire-electrode cutting along the rolling direction of the cold-rolled plate, and its thickness is 2 mm, as shown in Figure 1.

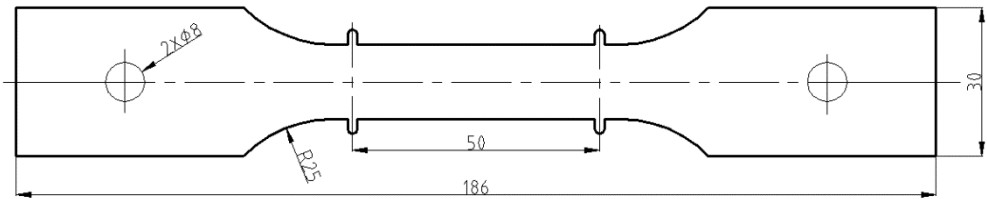

**Figure 1.** Dimension diagram of Al-Li-S4 aluminum alloy specimen (mm).

## 2.2. Creep Aging Test

The creep aging test machine adopts RDL50® type creep test machine provided by SUST Company (Zhuhai, China). The technological conditions of the test were as follows: firstly, the temperature was raised to 153 °C, the final temperature difference was ±1 °C at this temperature for 10 min, then the constant rate loading was carried out until the target load, and the applied stress was 200 MPa, 220 MPa and 240 MPa, respectively. After creep-aging test, the applied loading was released and the sample was naturally cooled down to the room temperature in the furnace. The intervals were 5, 10, 15 and 25 h.

## 2.3. Tensile Test

Tensile tests were performed using an SUST-CMT5105 machine (SUST Company, Zhuhai, China) at room temperature with a strain rate of 0.033 mm/s. All the surfaces of the samples were wet sanded before the tensile tests. The mechanical properties of the creep-aged alloys were evaluated according to national standard GBT228-2002. The maximum force after the yield stage is read from the recorded engineering stress and strain curve, and the tensile strength can be obtained by dividing the maximum force by the original cross-sectional area of the specimen. In addition, the stress at the specified residual elongation of 0.2% is taken as the yield strength of the material. In this paper, the elongation refers to the percentage elongation after fracture, which is calculated by dividing the permanent elongation (i.e., final gauge length minus original gauge length) by the original gauge length (50 mm). The origin gauge length of 50 mm is marked with a pan and the final length is measured by a micrometer with an accuracy of 0.02 mm. Each tensile test was repeated for at least two times.

## 2.4. Microstructure Observation and Characterization

Before observing the microstructure of the material, a dual-spray sample conforming to the shot was prepared. In this paper, a Titan G2 60-300 spherical aberration corrected projection electron microscope (FEI Company, Hillsboro, OR, USA) was used to capture the high-angle annular dark field (HAADF) inside the alloy, and the energy spectrum analysis (EDS) (FEI Company, Hillsboro, OR, USA) was performed. The instrument is operated at a high voltage of 300 kV, which can improve the imaging at higher magnification and further improve the observation accuracy of the structure of the precipitated phase and the accuracy of quantitative calculation.

## 3. Results and Discussion

### 3.1. Creep Behavior

Figure 2 shows the creep curves of Al-Li-S4 alloy under 200/220/240 MPa applied stress, respectively. As can be seen from Figure 2, there are two steady state creep stages in Al-Li-S4 alloy: the creep rate of the alloy decreases from the initial maximum value to a relatively balanced level, that is, the first steady-state creep; with the prolongation of aging time, the creep rate increases first and then decreases until it reaches a relatively balanced level. That is, the second stage of steady state creep. Table 2

shows the total creep ($\varepsilon_c$) and creep rates of Al-Li-S4 alloy at two steady-state stages ($\dot{\varepsilon}_{ss1}$ and $\dot{\varepsilon}_{ss2}$) under 200/220/240 MPa stress conditions.

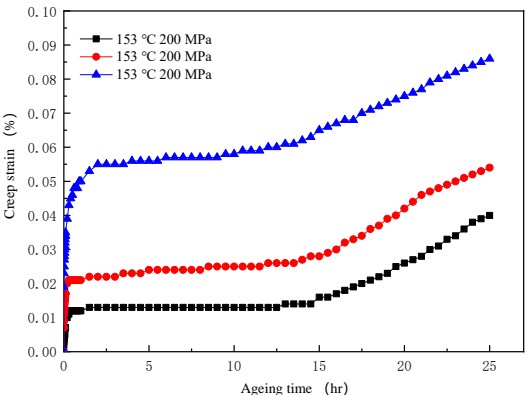

**Figure 2.** Creep curves of Al-Li-S4 aluminium alloy under different stress conditions.

**Table 2.** Total creep deformation ($\varepsilon_c$) and creep rates ($\dot{\varepsilon}_{ss1}$, $\dot{\varepsilon}_{ss2}$) at two steady-state stages of the Al-Li-S4 alloy under stress conditions of 200/220/240 MPa.

| Stress σ (MPa) | $\varepsilon_c$ (%) | $\dot{\varepsilon}_{ss1}$ | $\dot{\varepsilon}_{ss2}$ |
|---|---|---|---|
| 200 | 0.04 | $5.02 \times 10^{-5}$ | 0.00249 |
| 220 | 0.054 | $2.03 \times 10^{-4}$ | 0.00216 |
| 240 | 0.086 | $5.53 \times 10^{-4}$ | 0.00209 |

As shown in Figure 2 and Table 2, the creep of Al-Li alloys increases stepwise with aging time. For Al-Li alloys, due to the large amount of creep in the second stage of steady creep, the second stage of steady creep should be fully utilized in the creep aging process. Under different stress conditions, the creep of Al-Li-S4 alloy is different. The difference mainly comes from the initial creep stage, which indicates that stress plays a decisive role in the initial creep stage. With the increase of stress, the total creep of Al-Li alloy increases, which indicates that high stress can effectively improve the creep of Al-Li alloy. Moreover, the creep increment of Al-Li alloy shows a non-linear relationship with the increase of the same stress, because the yield strength of Al-Li-S4 alloy at 153 °C is about 225 MPa, the applied stress of 200 MPa and 220 MPa does not exceed the yield strength of the alloy, so the difference of creep strain between these two applied stress is about 0.014%. However, the applied stress of 240 MPa is greater than the yield strength at 153 °C, and plastic deformation occurs before the creep aging process, leading a large improvement in creep strain. Therefore, the difference of the creep strain between 220 and 240 MPa was larger than that between 200 and 240 MPa. The creep rate in the second steady state $\dot{\varepsilon}_{ss2}$ is generally greater than the creep rate in the first steady state $\dot{\varepsilon}_{ss1}$. However, with the increase of applied stress, $\dot{\varepsilon}_{ss1}$ gradually increases and $\dot{\varepsilon}_{ss2}$ gradually decreases, which indicates that the change of stress has a significant effect on creep rate.

It is well established by a number of researchers, including Lei et al. [16,17] and Li et al. [18,19], etc., that the power-law creep Equation, Equation (1), may be used to describe dislocation creep behavior. Therefore, the n value in Equation (2) can be derived from Equation (1), which is plotted (on a log–log axis) in Figure 3. Information of the stress exponent n can be used to verify the creep mechanisms [20].

$$\dot{\varepsilon}_s = A\sigma^n \exp\left(\frac{-Q}{RT}\right) \tag{1}$$

$$n = \frac{\ln \dot{\varepsilon}_s}{\ln \sigma} \tag{2}$$

As generally accepted: When $n = 1$ is the diffusion creep mechanism, grain boundary sliding is the main creep mechanism when $n = 2$; $n = 3$ is the dislocation creep mechanism; $n > 4$ is the dislocation creep mechanism. The higher the value of n, the faster the steady creep rate increases with the increase of stress, that is, the more sensitive it is to stress. Figure 3 shows that the stress exponents of the first and second stages of steady creep are different at 153 °C, which indicates that the creep mechanism has changed. The stress exponent $n_1 = 12$ in the first steady creep rate stage belongs to the dislocation climbing mechanism. It is noteworthy that the stress exponent $n_2 = -1 < 0$ of steady creep in the second stage is quite different from that of conventional aluminum alloys, which indicates that in the second stage of steady creep, the retardation of precipitation relative dislocation is greater than the enhancement of stress to dislocation, that is, precipitation phase plays a decisive role at this time. In view of this point, this paper will discuss in detail later.

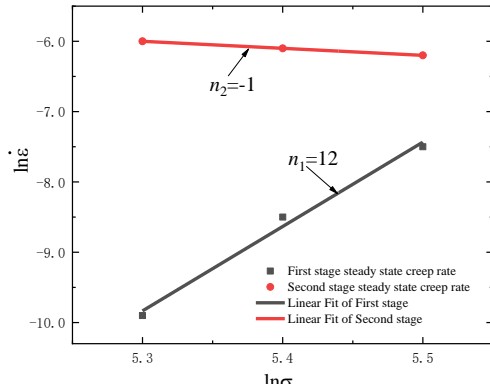

**Figure 3.** Stress exponent n of steady creep in two stages under different stress conditions at 153 °C.

*3.2. Evolution of Mechanical Properties*

Figure 4a represents the engineering stress–strain curves during the creep aging process at 155 °C and 240 MPa. With the increase of creep aging time, the flow stress increased, and the elongation decreased. Figure 4b refers to the stress–strain curves of 16 h creep-aged samples under the condition of 155 °C and different applied stresses. Also, the stress–strain curves of 16 h stress-free (0 MPa) aged sample are carried out. The 16 h creep-aged sample under the applied stress of 240 MPa has the highest flow stress. With the decrease of applied stress, the flow stress decreases. However, the flow stress of the stress-free aged sample is larger than that of creep-aged samples under the applied stresses of 200 and 220 MPa. In addition, the stress-free aged sample has the lowest elongation in the four aged samples with various applied stresses.

Figure 5 shows the tensile yield strength and elongation of Al-Li-S4 aluminum alloy after creep aging at 153 °C for 5, 10, 15, 20 and 25 h, respectively. It can be found from Figure 5a, the yield strength of the creep-aged sample under different applied stresses increases with the creep aging time. For instance, the yield strength was increased from 208 MPa for the initial material to 460 MPa after 25 h creep aging at 153 °C and 240 MPa. However, the increase rate of yield strength changed during the creep aging process. After the primary creep aging process (0–5 h), the yield strength increases by 32–72 MPa under different stress conditions. In the middle of the process (between 5 and 10 h), the yield strength increases slowly (about 10–15 MPa). Then, the yield strength improves significantly during the period of time between 10 and 25 h. The trend of elongation is opposite to that of yield strength during the creep aging process (Figure 5b). The elongation of the studied alloy decreases from 24% for initial material to 13% for 25 h creep-aged samples under different stresses of 200, 220 and 240 MPa. The elongation after stress-free aging is lower than that after creep aging; only about 9%. In Figure 5a; the yield strength increases with the increase of stress, which indicates that the stress level can promote the performance of components. However, when the aging time is 5–10 h, the increased rate of yield strength decreases, because the strengthening mechanism of the alloy is in

dynamic equilibrium. Under the combined action of dislocation strengthening, solution strengthening and precipitation strengthening, the increase rate of yield strength decreases. When the aging time is between 10 and 25 h, the yield strength increases continuously, which indicates that the material is still under-aged during this period. The yield strengths of the stress-free aged samples also show the multi-steps behavior, as plotted in Figure 5a. The yield strength under stress-free condition is higher than that under external stress condition at 15 h, which indicates that the external stress has restrained the aging precipitation process at 15 h, and this point stray out from other data at 0 MPa, because there are a lot of T$_1$ phase precipitation at 15 h. What's more, the yield strength tends to stable (known as peak aged state) after stress-free aging for 20 h, which indicates that the external stress can prolong the time of material reaching peak aged state. The peak yield strength under stress-free aging condition is lower than that under creep aging conditions, which indicates that the creep aging process can greatly improve the strength of materials and is beneficial to the realization of high strength of the formed components.

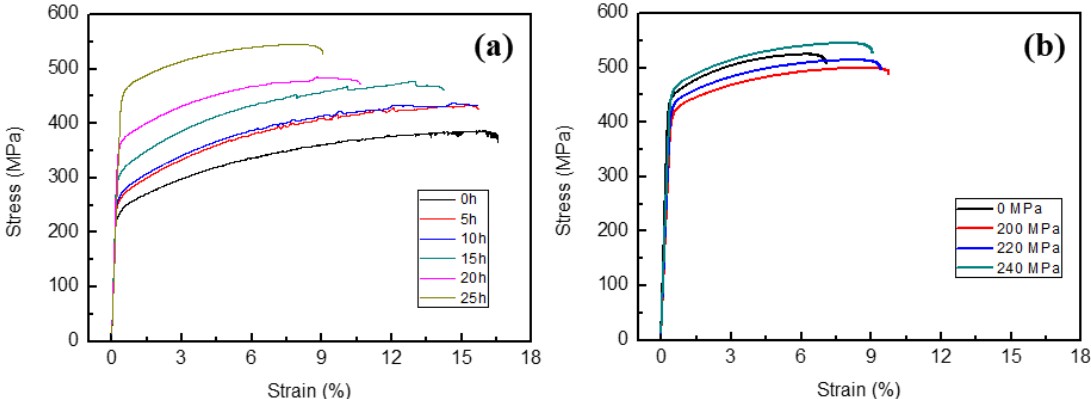

**Figure 4.** Engineering stress–strain curves of Al-Li-S4 alloy during the creep aging process at 155 °C and 240 MPa (**a**) and 16 h creep-aged samples under different applied stresses (**b**).

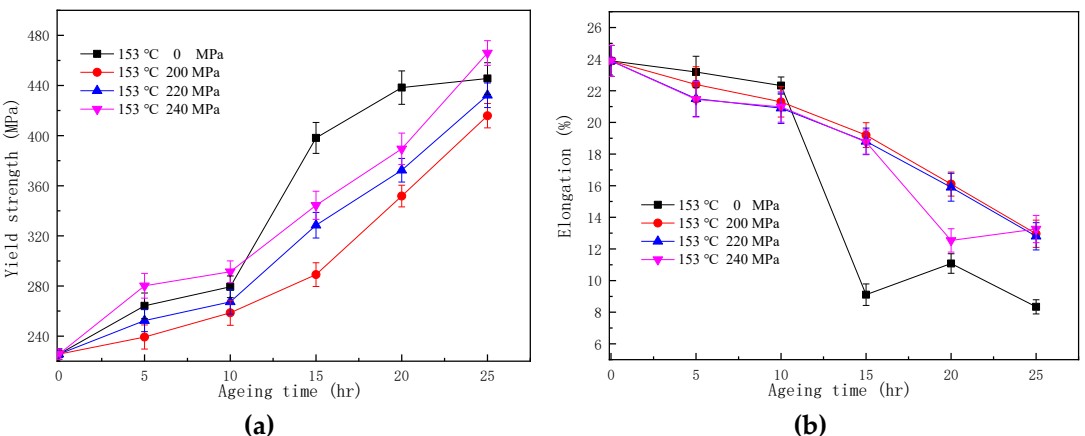

**Figure 5.** Mechanical properties of Al-Li-S4 aluminum alloy aged at 153 °C under different stresses, (**a**) Tensile yield strength curve, (**b**) Elongation curve.

### 3.3. Evolution of Microstructure

Figure 6 shows the HAADF-STEM of Al-Li-S4 alloy after creep aging. The electron beam direction is close to the crystal axis of [110] matrix. At the top left of each diagram, there is a corresponding relationship between the precipitation direction of T1 (Al2CuLi) and (Al2CuLi), Figure 6(a-1–d-1) the evolution of micro-precipitation for 1–25 h under 220 MPa applied stress, Figure 6(a-2–d-2) is a microscopic precipitation evolution diagram of 1–25 h under external stress of 240 MPa.

In the early stage of creep aging (1 h), there are a large number of dislocations induced by external stress in the alloy, which are also marked by red circles in the Figure 6(a-1). However, the dislocations in Figure 6(a-2) are obviously more than those in Figure 6(a-1). This indicates that the dislocations introduced by high stress are more than those in Figure 6(a-1). At the same time, there are a few precipitates in Figure 6(a-1,a-2), but they are not particularly obvious and are concealed under optical contrast. After ageing for 10 h, fine discoid/acicular T1 and θ' phase begin to precipitate unevenly, and there is little difference in the number ratio of the two precipitates. It can also be seen that both T1 and θ' phases precipitate along dislocation, which is consistent with the conclusion found in reference [21]. Moreover, the phase–orientation relationship of the two precipitates has been shown in the upper right corner of the graph, as shown in Figure 6(b-1,b-2). Aging lasted for 15 h, and a large number of $T_1$ phases appeared in Figure 6(c-1,c-2) with uniform distribution. Compared with aging for 5 h, the number of $T_1$ and θ' phases increased and the growth rate of $T_1$ phases was faster than θ' phases with time. Most $T_1$ and θ' phases relative matrix precipitated along (11-1)/(1-11) and (200) planes, respectively. From Figure 6(d-1,d-2), the distribution of $T_1$ phase is more uniform, the length and thickness of $T_1$ phase increase significantly, the number of θ' phases decrease, but the length and thickness increase.

The precipitated phases of Al-Li alloy mainly consist of $T_1$ phase and θ' phase after creep aging at 153 °C (Figure 6), but because the number of θ' phases is very small and the number of $T_1$ phase is much more than that of θ' phase, the latter analysis in this paper is mainly based on $T_1$ phase. Using Image-pro plus image processing software, the length distribution of $T_1$ phase was measured manually from the TEM image area (at least 200 precipitates in two sample areas), as shown in Figure 7. Comparing Figure 7a,c, the statistical averages are different, specifically, the statistical average of the data in Figure 6a is 75.79 nm; the statistical average of the data in Figure 7c is 120.5 nm. The peak value of $T_1$ phase length distribution in stress-free aging shifts to the right, indicating that the $T_1$ phase grows gradually with aging time increasing. The statistical average of the data in Figure 7b is 107.48 nm; the statistical average of the data in Figure 7d is 81.09 nm. The peak value of $T_1$ phase length distribution of 220 MPa stress aging shifted slightly to the right, indicating that $T_1$ phase did not continue to grow during this process. And the distribution of Figure 7c is more uniform than that of Figure 7d, which indicates that the distribution of $T_1$ phase is more uniform than that of stress-free aging.

According to reference [22], the relative volume fraction is used to express the change of the number of precipitated phases, as shown in Equation (3).

$$\overline{f_v} = f_v / f_P \tag{3}$$

where $f_v$ is the true volume fraction and $f_P$ is the maximum volume fraction. Due to the actual application requirements of the project and considering the manufacturing cost, the creep aging process of the alloy Al-Li-S4 aluminum alloy was not subjected to peak aging, only to 25 h. Therefore, in this paper, the maximum volume fraction of precipitated phases at different time in actual measurement is taken. At the same time, because Al-Li-S4 aluminum alloy has multiphase precipitation, in order to clarify the precipitation competition among various precipitated phases, the sum of T1 and phase volume fraction is taken as the maximum value. Figure 8 shows the relative volume fraction of T1 phase and phase of Al-Li-S4 aluminum alloy under different stress creep aging conditions. From Figure 8, it can be seen that the relative volume fraction of $T_1$ phase and phase did not change at 15–25 h, indicating that $T_1$ phase and phase did not precipitate after 15 h and began to coarsen.

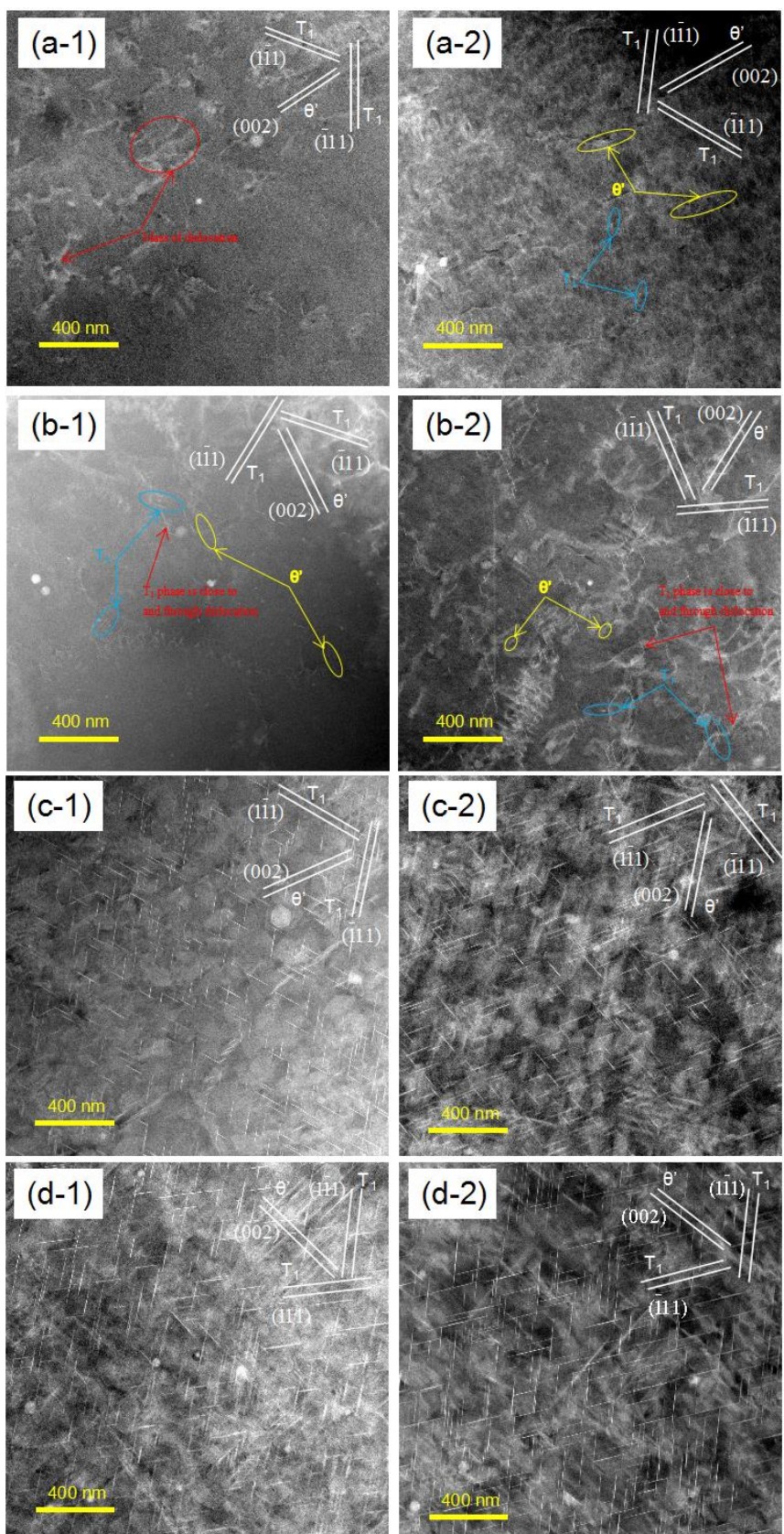

**Figure 6.** HAADF-STEM micro-images of Al-Li-S4 Al alloy during creep aging at 153 °C, T1 and phase evolution with time, electron beam direction approaching [110] matrix crystal axis direction; 220 MPa applied stress (**a-1–d-1**): 1 h, 10 h, 15 h, 25 h; 240 MPa applied stress (**a-2–d-2**): 1 h, 10 h, 15 h, 25 h.

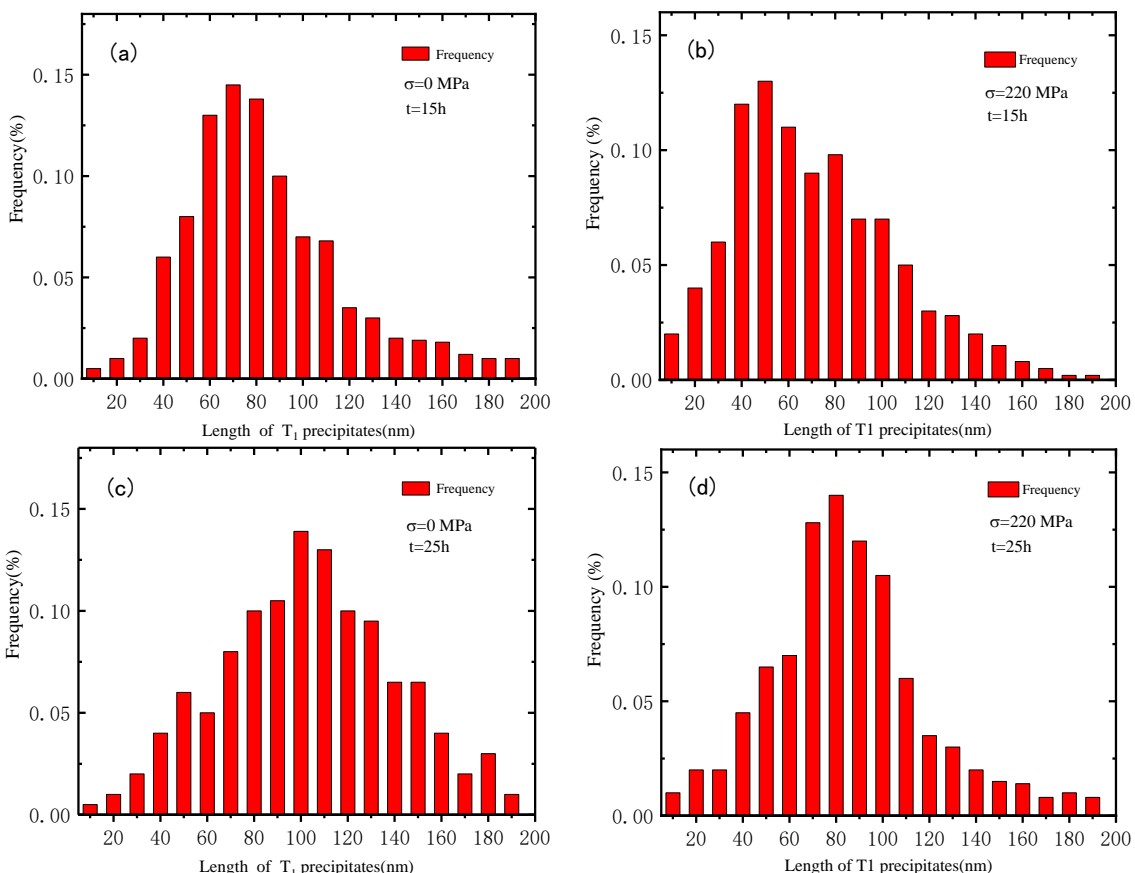

**Figure 7.** Diameter distribution of T1 phase of Al-Li-S4 aluminum alloy after creep aging at 153 ((**a**) σ = 0, t = 15 h; (**b**) σ = 220 MPa, t = 15 h; (**c**) σ = 0, t = 25 h; (**d**) σ = 220 MPa, t = 25 h).

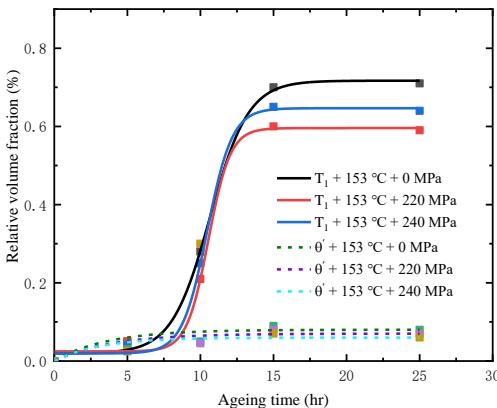

**Figure 8.** The relative volume fraction curves of $T_1$ phase and phase of Al-Li-S4 aluminum alloy under different stress creep aging conditions.

## 4. Discussion

### 4.1. Multi-Step Creep Aging Behavior

Figure 9 shows the division of creep curves with different stages of aging process. The creep curves can be divided into four stages.

In the first stage of creep, although the solute elements in the matrix are consumed due to the partial precipitation of precipitated phase, the creep rate increases sharply due to the application of applied stress and the continuous movement of dislocations, and dislocation strengthening plays

a major role in this stage, which leads to the decrease of creep rate. The larger the applied stress is, the more dislocations are introduced, and the more the creep amount produced in the first stage of creep is. This indicates that the difference caused by different applied stress mainly occurs in the first stage of creep. In the second stage of creep (steady state creep in the first stage), $T_1$ and $\theta'$ phase precipitate continuously at dislocations, solute elements in the matrix continue to be consumed, dislocations decrease, and the equilibrium state between precipitated phase, solute atoms and dislocations is reached. In the third stage of creep, it can be seen that due to the accelerated nucleation and precipitation of $T_1$ phase, the number of solute atoms in the matrix decreases sharply, which leads to the decrease of creep resistance [22,23], and the increase of creep rate at this stage. In the fourth stage of creep (steady state creep in the second stage), as aging proceeds, $T_1$ phase begins to coarsen, which makes dislocations more prone to occur and causes creep to greatly increase, but the coarsened $T_1$ phase distributes more uniformly and compactly under stress.

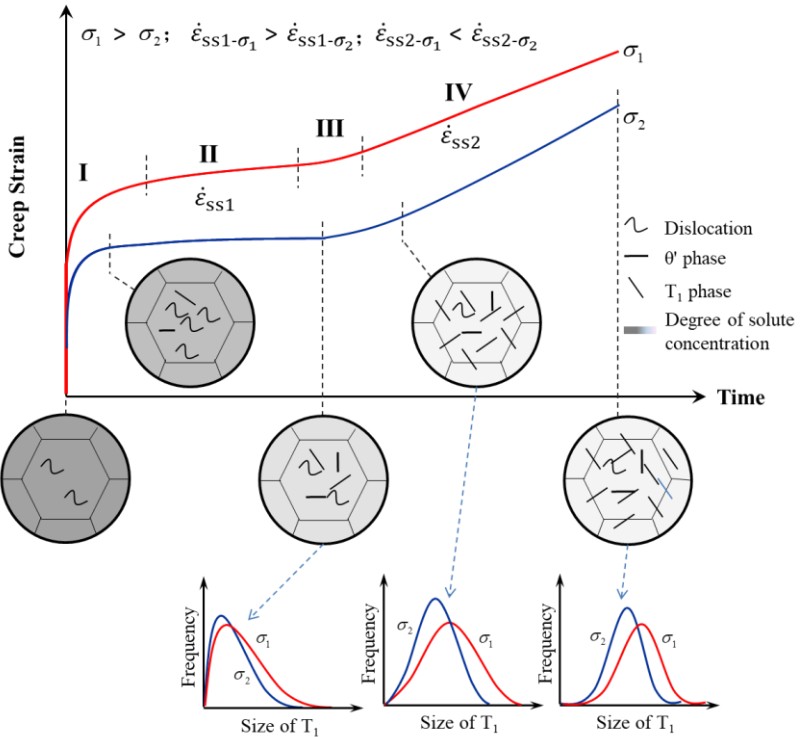

**Figure 9.** Schematic diagrams of creep dominant factors at different stages in the "multi-step" creep curve of Al-Li-S4 aluminum alloy.

*4.2. The Multi-Step Phenomenon of Mechanical Properties*

On the basis of the experimental results, a schematic diagram can be supposed for considering the contribution of the different acting mechanisms to the yield strength, which is shown in Figure 10. The multi-step of the studied Al-Li alloy in yield strength can be divided by procedure into four stages, that is, rapid increase stage (caused by strain hardening), slow increase stage, second rapid increase stage (caused by precipitation hardening) and slow decrease stage (known as over aged stage). For the change in the yield strength of the alloy in Figure 5a, first, the relationship between the strength of the alloy and the precipitated phase is such that the size and distribution of the precipitated phase determine the strength of the alloy. According to the change of the precipitated phase described above, the $T_1$ phase is the main strengthening phase of the alloy and tends to be stable after 15 h and begins to coarsen.

In the first rapid increase stage, the yield strength of the alloy increases with an increase of time, because strain hardening occurs at this stage. In the slow increase stage, the rate of increase in yield strength decreases, because during this stage, the contribution of dislocation strengthening and solid

solution strengthening to the strength of the alloy is gradually weakened, while the strengthening effect of the precipitate is gradually increasing, and it is possible that these strengthening mechanisms reach a state of dynamic equilibrium between 5 and 10 h. In the second rapid increase stage, the yield strength of the alloy increases rapidly, since with the increase of aging time, the $T_1$ phase is getting more and more, which resulting in a rapid increase in strength. This stage belongs to precipitation hardening. In the slow decrease stage, with the increase of aging time, this stage is the over-aging stage, and the yield strength decreases.

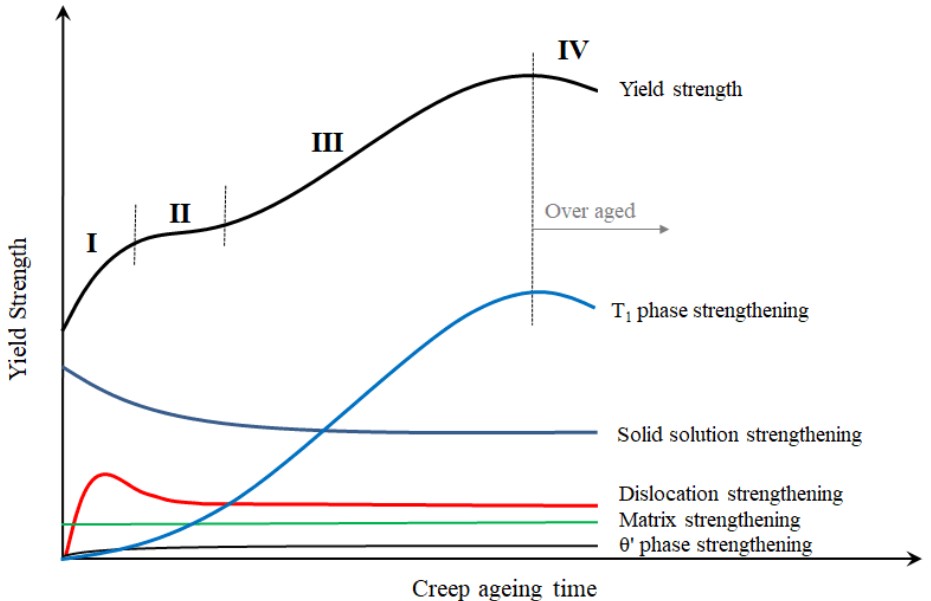

**Figure 10.** Schematic diagram of the supposed contribution of each strengthening mechanism to the yield strength of Al-Li-S4 alloy.

After 15 h, the $T_1$ phase begins to coarsen, and the strength of the alloy still increases with the increase of aging time, this is because after 15 h, the distribution of the $T_1$ phase becomes more uniform and finer (Figure 6d) under the applied stress. When the aging time is between 10–25 h, the yield strength under stress-free conditions is higher than that under applied stress conditions, since the applied stress has an inhibitory effect to the growth of the $T_1$ phase. The yield strength of the stress-free aging process is advanced to the peak than the stress aging process, because under stress-free conditions, the original coarse $T_1$ phase absorbs the surrounding Cu atoms and continues to grow, moreover, due to the suppression of applied stress, the size of the $T_1$ phase is smaller than the size of the $T_1$ phase under stress-free conditions, which represents higher strength. The strength of stress-free aging for 25 h is greater than the strength of stress aging of 200 and 220 MPa, but less than the stress aging of 240 MPa. This is because the greater the applied stress, the more obvious the inhibition of the growth of $T_1$ phase, the more uniform the distribution of $T_1$ phase. The macro aspect is the higher the intensity.

## 5. Conclusions

It will probably be a trend to manufacture large Al-Li alloy components by creep age forming technology. Thus, the creep aging behavior of Al-Li-S4 alloy was studied. The main findings are as follows:

(1)    Creep behavior presents double steady state creep stages during the creep aging process of Al-Li-S4 alloy. With the increase of stress level, the first steady creep rate increased, but the second steady creep rate was slightly reduced. The change in the stress exponent, *n*, shows that the mechanism of the first steady creep stage and the second steady creep stage are different. Moreover, the creep

strain change in the second steady creep stage is mainly related to the change of the number and size of precipitates under different applied stress. It indicated that there is a strong interaction between stress-dependent precipitates and creep deformation.

(2) Mechanical properties of Al-Li-S4 alloy also show multi-step behaviors during the creep aging process. Although the yield strength has been increasing, the increasing rate is different. The increasing rate decreases first and is then followed by typical age strengthening rules. It suggested that the primary stage of yield strength is correlate to the strain strengthening induced by creep of Al-Li-S4 alloy.

(3) In the creep aging process, Al-Li-S4 alloy mainly contains two precipitation phases, $T_1$ phase and $\theta'$ phase. Among them, $T_1$ phase is the main strengthening phase of the studied alloy. $T_1$ phase has a long incubation period, which is longer than that time of primary creep stage caused by dislocation increment. With the increase of stress level, the size and distribution of precipitates are more uniform, which is the main reason for the increase of mechanical properties under high stress conditions.

**Author Contributions:** Y.M. has performed the creep and TEM experiments, the theoretical explanation of experimental curves; F.X. has performed the statistics and processing of experimental data; L.Z. has performed the design of the work; Y.X. has performed the discussion of the multi-step phenomenon and final approval of the version to be published.

**Funding:** This work was supported by the National key R&D Program of China (No. 2017YFB0306300); National Natural Science Foundation of China (No. 51675538, 51601060); National Defense Program of China (No. JCKY2014203A001); Free Exploration Project of State Key Laboratory of High-performance Complex Manufacturing (No. ZZYJKT2019-11).

**Acknowledgments:** Authors would like to thank Southwest Aluminum Industry (Group) Co., Ltd. for support in materials used for experiments.

**Conflicts of Interest:** The authors declare no conflict of interest.

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
