# Peer review of "Study on Multi-Step Creep Aging Behavior of Al-Li-S4 Alloy"

_metals, doi:10.3390/met9070807_

Reviewer 1 Report

The authors rewrote the section 2-2 "Tensile test". I think corrections were accomplished.  So I decide it is acceptable.

Author Response

Thank you for your comments on this paper.

Reviewer 2 Report

The authors have finally addressed my concerns in the previous manuscript. I support the publication of this revised manuscript in Metals. The authors should do some minor editing to  correct typo and grammatical errors. 

Author Response

Typo and grammatical errors have been corrected, and thank you for your comments on this paper.

Reviewer 3 Report

No further comment.

Author Response

Thank you for your comments on this paper.

This manuscript is a resubmission of an earlier submission. The following is a list of the peer review reports and author responses from that submission.

Round  1

Reviewer 1 Report

The authors investigated the multi-step creep aging behavior of Al-Li-2 S4 alloys. The results are interesting.

Figure 2 : Explain the reason why the difference of the creep strain between 220 MPa and 240 MPa was larger than that between 200 MPa and 240 MPa.

Figure 4 : Explain the reason why the data at 0 MPa and 15 hr stray out from other data at 0 MPa.

Figure 6 : When you found the mean value statistically, is the difference of the mean value recognized?

L326-L327 : "It indicated that there is a strong interaction between stress dependent precipitates and creep deformation."  Please explain the interaction more definitely and in detail.

Reference lists : Please repair it to a form of "metals".

Author Response

Response to Reviewer 1 Comments

Point 1: Figure 2 : Explain the reason why the difference of the creep strain between 220 MPa and 240 MPa was larger than that between 200 MPa and 240 MPa.

Response 1:Because the yield strength of Al-Li-S4 alloy at 153 °C is about 225 MPa, the applied stress of 200 MPa and 220 MPa does not exceed the yield strength of the alloy, so the difference of creep strain between these two applied stress is about 0.014%. However the applied stress of 240MPa is greater than the yield strength at 153 °C, and plastic deformation occurs before the creep aging process, leading a large improvement in creep strain. Therefore, the difference of the creep strain between 220 MPa and 240 MPa was larger than that between 200 MPa and 240 MPa.

Point 2: Figure 4 : Explain the reason why the data at 0 MPa and 15 hr stray out from other data at 0 MPa.

Response2: For artificial aging, within 10 hours, the yield strength of the alloy increases with time, but between 10 and 15 h, the increment is larger, which looks different from other points,the yield strength of the alloy is basically stable from 20 h to 25 h; Moreover, the yield strength under stress-free condition is higher than that under external stress condition at 15h, which indicates that the external stress has restrained the aging precipitation process at 15h, and this point stray out from other data at 0 MPa, because there are a lot of T1 phase precipitation at 15h.

Point 3: Figure 6 : When you found the mean value statistically, is the difference of the mean value recognized?

Response 3:  Comparing Fig. 6 (a) and Fig. 6 (c), the statistical averages are different, specifically, the statistical average of the data in Fig. 6 (a) is 75.79nm; the statistical average of the data in Fig. 6 (c) is 120.5nm. The peak value of T1 phase length distribution in stress-free aging shifts to the right, indicating that the T1 phase grows gradually with aging time increasing. In Fig. 6 (b) and Fig. 6 (d),specifically, the statistical average of the data in Fig. 6 (b) is 107.48nm; the statistical average of the data in Fig. 6 (d) is 81.09nm. The peak value of T1 phase length distribution of 220 MPa stress aging shifted slightly to the right, indicating that T1 phase did not continue to grow during this process.

Point 1: L326-L327 : "It indicated that there is a strong interaction between stress dependent precipitates and creep deformation."  Please explain the interaction more definitely and in detail.

Response 4: Accroding to the change in the stress exponent n, it shows that the mechanism of the first steady creep stage and the second steady creep stage are different. Moreover, the creep strain change in the second steady creep stage is mainly relate to the change of the number and size of precipitates under different applied stress.

Reviewer 2 Report

This  paper fits the scope of metals. However, it can not support its publications in the current form for the following reasons:

It has lots of grammatical  and typo errors and therefore the entire manuscripts must be edited and proofread. For example in the Abstract, using the words "making" and "rate" on lines 22 and 27 respectively does not make grammatical sense. There are lots of strange words also in the paper such as "emetal" and "efor".

What does the author mean by "rebound of components" in line 60?

Section 2.3 on Macro Performance Testing of Materials" is not clear to the reviewer at all. Specifically, what does the author mean by "calibrating the calibration distance"?  Also, How can micrometer be used to measure a mechanical property such as the yield strength? What does the author also mean by tensile strength at the end of the tension. This portion looks strange. To the reviewer's knowledge, no instrument is used to directly measure stress. 

Some of the claims in Section 3.2. Evolution of Mechanical Properties do not make sense or even contradictory. For example, in paragraph 1,  the author claimed that the yield strength is stable from 10 h to 15 h and  decrease from 20 h to 25 h. These claims do not agree with the figure and more so the author made another contradictory claim regarding the yield strength on line 171. This section is poorly written and must be re-written.

Author Response

Response to Reviewer 2 Comments

Point 1: It has lots of grammatical  and typo errors and therefore the entire manuscripts must be edited and proofread. For example in the Abstract, using the words "making" and "rate" on lines 22 and 27 respectively does not make grammatical sense. There are lots of strange words also in the paper such as "emetal" and "efor".

Response 1:The grammatical problem has been revised. 1.At the same time, the increase of stress level effectively inhibits the growth of T1 phase, which resulting in these strengthening phases more uniform, and thus improves the mechanical properties of materials. 2.It is considered that the main reasons for the multi-step phenomenon of creep and mechanical properties are strongly relate to the nucleation, growth and distribution of T1 phase. 3.Creep Ageing Forming (CAF) is a metal forming technology developed efor manufacturing large aluminum components in aerospace industry [1]. In recent years, Al-Li alloys have attracted much attention because of low weight,high stiffness and high strength for their applications in aircraft, aerospace and military fields [2-5].

Point 2: What does the author mean by "rebound of components" in line 60?

Response2: Experiments show that the stress relaxation becomes more obvious with the increase of temperature, and the rebound of components decreases,which means that he forming accuracy of the component is improved.

Point 3: Section 2.3 on Macro Performance Testing of Materials" is not clear to the reviewer at all. Specifically, what does the author mean by "calibrating the calibration distance"?  Also, How can micrometer be used to measure a mechanical property such as the yield strength? What does the author also mean by tensile strength at the end of the tension. This portion looks strange. To the reviewer's knowledge, no instrument is used to directly measure stress. 

Response 3:Tensile test of creep aged samples were completed on CMT-5105 electronic universal testing machine with the strain rate of 2 mm/min at room temperature. In order to measure the elongation of the sample more accurately, the gauge length is calibrated before the tests, and measured again at the end of the tests. Three repeated tests were carried out under the same experimental conditions, and the average of results was taken as the final performance for comparative analysis.

Point 4: Some of the claims in Section 3.2. Evolution of Mechanical Properties do not make sense or even contradictory. For example, in paragraph 1,  the author claimed that the yield strength is stable from 10 h to 15 h and  decrease from 20 h to 25 h. These claims do not agree with the figure and more so the author made another contradictory claim regarding the yield strength on line 171. This section is poorly written and must be re-written.

Response 4:  Fig. 4 shows the tensile yield strength and elongation of Al-Li-S4 aluminum alloy after creep aging tests at 153℃, 0/200/220/240 MPa and 5 h, 10 h, 15 h, 20 h and 25 h, respectively. In Fig. 4 (a), the yield strength of the alloy increases with aging time as a whole; for creep aging, the yield strength of the alloy increases with time within 10 hours, and the increment is very small. After 10 hours, the yield strength of the alloy increases linearly with time, and with the increase of applied stress (200-240MPa), the yield strength of the alloy increases almost linearly. For artificial aging, within 10 hours, the yield strength of the alloy increases with time, but between 10 and 15 h, the increment is larger, which looks different from other points,the yield strength of the alloy is basically stable from 20 h to 25 h;

Reviewer 3 Report

line 157: explain that the Fig.4 results have been obtained at room temperature or specify if other.

line 162-165: these sentences are confusing, please use shorter and clear sentences. 

line 181: indicate in the Fig.4 label that the mecanical properties are related to room temperature

line 301-302: it is strongly suggested to use the following forms: "On the base of the experimental results a schematic diagram can be supposed for taking into account of the contribution of the different acting mechanism to the yield strength …..", please continue and arrange as you prefer.

line 304: Figure 9 caption: "schematic diagram of the supposed contribution of each…."

Author Response

Response to Reviewer 3 Comments

Point 1: line 157: explain that the Fig.4 results have been obtained at room temperature or specify if other.

Response 1:The Fig.4 results have been obtained at 153 ℃.

Point 2: line 162-165: these sentences are confusing, please use shorter and clear sentences. 

Response2:  It is noteworthy that the stress exponent n2=-1<0 of steady creep in the second stage is quite different from that of conventional aluminium alloys, which indicates that in the second stage of steady creep, the precipitation phase plays a decisive role at this time. In view of this point, this paper will discuss in detail later.

Point 3:  indicate in the Fig.4 label that the mechanical properties are related to room temperature

Response 3: The mechanical properties in the Fig.4 is obtained at  153 ℃.

Point 4:line 301-302: it is strongly suggested to use the following forms: "On the base of the experimental results a schematic diagram can be supposed for taking into account of the contribution of the different acting mechanism to the yield strength …..", please continue and arrange as you prefer.

Response 4:  On the base of the experimental results, a schematic diagram can be supposed for taking into account of the contribution of the different acting mechanism to the yield strength, which is shown in Fig.9.

Point 5:line 304: Figure 9 caption: "schematic diagram of the supposed contribution of each…."

Response 5:  Figure 9. Schematic diagram of the supposed contribution of each strengthening mechanism to the yield strength of Al-Li-S4 alloy.

Round  2

Reviewer 2 Report

wer's Comments

The reviewer still do not support the publication of this paper in the current form. It appears the revisions were done in a hurry and did not thoroughly address the reviewer's initial concerns as follow

1. There are still significant grammatical errors in paper some of which were pointed out in the initial review but were never addressed. For example the word "relate" on line 27 is not grammatically correct when preceded by the word "are". Also strange words line  "efor"  on line 33 and some others in other parts of the manuscript are still left uncorrected even after they had been pointed out in the initial review. It appears the whole corrections and proofreading were done in a hurry.

2.  The authors responses to the reviewer's concern on Section 2.3 on Macro Performance Testing of Materials"  are completely inappropriate. Specifically, what does the authors mean by "calibrating the gage length"? Also, the issue of how a micrometer can be used to measure a mechanical property such as the yield strength was completely ignored and deleted from the revised manuscript rather than being addressed. The authors did not even state how they measure the mechanical properties such as the yield strength  in the revised manuscript. This is completely unacceptable. The results of the tensile test stress-strain curves should also be presented. 

3: Some of the claims in Section 3.2. still do not agree with the graphs presented.  For example the reviewer did not see any stable yield strength for any of the given time interval as claimed by the authors. The discussion presented in this section looks like a mere technical report.

Author Response

Response to  Reviewer 2 Comments

Point 1: There are still significant grammatical errors in paper some of which were pointed out in the initial review but were never addressed. For example the word "relate" on line 27 is not grammatically correct when preceded by the word "are". Also strange words line  "efor"  on line 33 and some others in other parts of the manuscript are still left uncorrected even after they had been pointed out in the initial review. It appears the whole corrections and proofreading were done in a hurry. 

Response 1: The English and grammar mistakes have been corrected carefully throughout the manuscript.

Point 2: The authors responses to the reviewer's concern on Section 2.3 on Macro Performance Testing of Materials"  are completely inappropriate. Specifically, what does the authors mean by "calibrating the gage length"? Also, the issue of how a micrometer can be used to measure a mechanical property such as the yield strength was completely ignored and deleted from the revised manuscript rather than being addressed. The authors did not even state how they measure the mechanical properties such as the yield strength  in the revised manuscript. This is completely unacceptable. The results of the tensile test stress-strain curves should also be presented. 

Response2: The section 2.3 has been rewritten. The mechanical properties of the creep aged alloys was evaluated according to national standard GBT228-2002.

I'm sorry that we can't provide the tensile test stress-strain curves, but we guarantee the authenticity and repeatability of the data.

Point 3: Some of the claims in Section 3.2. still do not agree with the graphs presented.  For example the reviewer did not see any stable yield strength for any of the given time interval as claimed by the authors. The discussion presented in this section looks like a mere technical report.

Response 3: The section 3.2 has been rewritten. Some of the claims in Section 3.2 have been improved. The description of “stable yield strength” has been changed to “slow increasing yield strength”. The multi-step of the studied Al-Li alloy in yield strength can be divided by procedure into four stages, that is, first rapid increase stage (caused by strain hardening), slow increase stage, second rapid increase stage (caused by precipitation hardening) and slow decrease stage (known as over aged stage). The discussion of multi-step phenomenon in yield strength has been presented in the section 4.2.

Round  3

Reviewer 2 Report

It appears the authors did not want to respond to important technical issues raised in my previous evaluations. For this reason, the paper should be rejected. 

More specifically, the  issues raised such as what does the authors mean by "calibrating the gage length in the original paper was never addressed but deleted"? Also, the issue of how a micrometer can be used to measure a mechanical property such as the yield strength had been completely ignored and deleted from the two previous revised manuscript rather than being addressed. I also asked the authors to state how they measured  the  mechanical properties such as the yield strength but rather than doing that they are quoting a national standard. The authors' response that they cannot provide the tensile stress-strain curves and claiming authenticity of the data is very ridiculous. If the stress-strain curves cannot be provided and how the measurements were done cannot be described, then the paper itself is questionable and should therefore be rejected.